# PanRep: Universal node embeddings for heterogeneous graphs

## Abstract

Learning unsupervised node embeddings facilitates several downstream tasks such as node classification and link prediction. A node embedding is universal if it is designed to be used by and benefit various downstream tasks. This work introduces PanRep, a graph neural network (GNN) model, for unsupervised learning of universal node representations for heterogenous graphs. PanRep consists of a GNN encoder that obtains node embeddings and four decoders, each capturing different topological and node feature properties. Abiding to these properties the novel unsupervised framework learns universal embeddings applicable to different downstream tasks. PanRep can be furthered fine-tuned to account for possible limited labels. In this operational setting PanRep is considered as a pretrained model for extracting node embeddings of heterogenous graph data. PanRep outperforms all unsupervised and certain supervised methods in node classification and link prediction, especially when the labeled data for the supervised methods is small. PanRep-FT (with fine-tuning) outperforms all other supervised approaches, which corroborates the merits of pretraining models. Finally, we apply PanRep-FT for discovering novel drugs for Covid-19. We showcase the advantage of universal embeddings in drug repurposing and identify several drugs used in clinical trials as possible drug candidates.

## 1 Introduction

Learning node representations from heterogeneous graph data powers the success of many downstream machine learning tasks such as node classification (Kipf & Welling, 2017), and link prediction (Wang et al., 2017). Graph neural networks (GNNs) learn node embeddings by applying a sequence of nonlinear operations parametrized by the graph adjacency matrix and achieve state-of-the-art performance in the aforementioned downstream tasks. The era of big data provides an opportunity for machine learning methods to harness large datasets (Wu et al., 2013). Nevertheless, typically the labels in these datasets are scarce due to either lack of information or increased labeling costs (Bengio et al., 2012). The lack of labeled data points hinders the performance of supervised algorithms, which may not generalize well to unseen data and motivates *unsupervised* learning.

Unsupervised node embeddings may be used for downstream learning tasks, while the specific tasks are typically not known a priori. For example, node representations of the Amazon book graph can be employed for recommending new books as well as classifying a book's genre. This work aspires to provide *universal* node embeddings, which will be applied in multiple downstream tasks and achieve comparable performance to their supervised counterparts.

Although unsupervised learning has numerous applications, limited labels of the downstream task may be available. Refining the unsupervised universal representations with these labels could further increase the representation power of the embeddings. This can be achieved by *fine-tuning* the unsupervised model. Natural language processing methods have achieved state-of-the-art performance by applying such a fine-tuning framework (Devlin et al., 2018). Fine-tuning pretrained models is beneficial compared to end-to-end supervised learning since the former typically generalizes better especially when labeled data are limited and decreases the inference time since typically just a few fine-tuning iterations typically suffice for the model to converge (Erhan et al., 2010).

This work introduces a framework for unsupervised learning of universal node representations on heterogenous graphs termed PanRep[1]. It consists of a GNN encoder that maps the heterogenous graph data to node embeddings and four decoders, each capturing different topological and node feature

---

[1]Pan: Pangkosmios (Greek for universal) and Rep: Representation

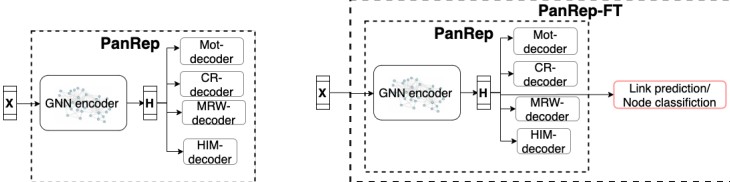

Figure 1: Illustration of the PanRep (left) and PanRep-FT (right) models. The GNN encoder processes the node features $\mathbf{X}$ to obtain the embeddings $\mathbf{H}$. The decoders facilitate unsupervised learning of $\mathbf{H}$.

properties. The cluster and recover (CR) decoder exploits a clustering prior of the node attributes. The motif (Mot) decoder captures structural node properties that are encoded in the network motifs. The meta-path random walk (MRW) decoder promotes embedding similarity among nodes participating in a MRW and hence captures intermediate neighborhood structure. Finally, the heterogeneous information maximization (HIM) decoder aims at maximizing the mutual information among node local and the global representations per node type. These decoders model general properties of the graph data related to node homophily (Gleich, 2015; Kloster & Gleich, 2014) or node structural similarity (Rossi & Ahmed, 2014; Donnat et al., 2018). PanRep is solely supervised by the decoders and has no knowledge of the labels of the downstream task. The universal embeddings learned by PanRep are employed as features by models such as SVM (Suykens & Vandewalle, 1999) or DistMult (Yang et al., 2014) to be trained for the downstream tasks. To further accommodate the case where limited labels are available for some downstream tasks we propose fine-tuning PanRep (PanRep-FT). In this operational setting, PanRep-FT is optimized adhering to a task-specific loss. PanRep can be considered as a pretrained model for extracting node embeddings of heterogenous graph data. Figure 1 illustrates the two novel models. The contribution of this work is threefold.

**C1.** We introduce a novel problem formulation of universal unsupervised learning and design a tailored learning framework termed PanRep. We identify the following general properties of the heterogenous graph data: (i) the clustering of local node features, (ii) structural similarity among nodes, (iii) the local and intermediate neighborhood structure, (iv) and the mutual information among same-type nodes. We develop four novel decoders to model the aforementioned properties.

**C2.** We adjust the unsupervised universal learning framework to account for possible limited labels of the downstream task. PanRep-FT refines the universal embeddings and increases the model generalization capability.

**C3.** We compare the proposed models to state-of-the-art supervised and unsupervised methods for node classification and link prediction. PanRep outperforms all unsupervised and certain supervised methods in node classification, especially when the labeled data for the supervised methods is small. PanRep-FT outperforms even supervised approaches in node classification and link prediction, which corroborates the merits of pretraining models. Finally, we apply our method on the drug-repurposing knowledge graph (DRKG) for discovering drugs for Covid-19 and identify several drugs used in clinical trials as possible drug candidates.

## 2 RELATED WORK

**Unsupervised learning.** Representation learning amounts to mapping nodes in an embedding space where the graph topological information and structure is preserved (Hamilton et al., 2017). Typically, representation learning methods follow the encoder-decoder framework advocated by PanRep. Nevertheless, the decoder is typically attuned to a single task based on e.g., matrix factorization (Tang et al., 2015; Ahmed et al., 2013; Cao et al., 2015; Ou et al., 2016), random walks (Grover & Leskovec, 2016; Perozzi et al., 2014), or kernels on graphs (Smola & Kondor, 2003). Recently, methods relying on GNNs are increasingly popular for representation learning tasks (Wu et al., 2020). GNNs typically rely on random walk-based objectives (Grover & Leskovec, 2016; Hamilton et al., 2017) or on maximizing the mutual information among node representations (Veličković et al., 2018b). Relational GNNs methods extend representation learning to heterogeneous graphs (Dong et al., 2017; Shi et al., 2018; Shang et al., 2016). Relative to these contemporary works PanRep introduces multiple decoders to learn universal embeddings for heterogeneous graph data capturing the clustering of local node features, structural similarity among nodes, the local and intermediate neighborhood structure, and the mutual information among same-type nodes.

**Supervised learning.** Node classification is typically formulated as a semi-supervised learning (SSL) task over graphs, where the labels for a subset of nodes are available for training (Belkin et al., 2004). GNNs achieve state-of-the-art performance in SSL by utilizing regular graph convolution (Kipf & Welling, 2017) or graph attention (Veličković et al., 2018a), while these models have been extended in heterogeneous graphs (Schlichtkrull et al., 2018; Fu et al., 2020; Wang et al., 2019b). Similarly, another prominent supervised downstream learning task is link prediction with numerous applications in recommendation systems (Wang et al., 2017) and drug discovery (Zhou et al., 2020; Ioannidis et al., 2020). Knowledge-graph (KG) embedding models rely on mapping the nodes and edges of the KG to a vector space by maximizing a score function for existing KG edges (Wang et al., 2017; Yang et al., 2014; Zheng et al., 2020). RGCN models (Schlichtkrull et al., 2018) have been successful in link prediction and contrary to KG embedding models can further utilize node features. The universal embeddings extracted from PanRep without labeled supervision offer a strong competitive to these supervised approaches for both node classification and link prediction tasks.

**Pretraining.** Pretraining models provides a significant performance boost compared to traditional approaches in natural language processing (Devlin et al., 2018; Radford et al., 2019) and computer vision (Donahue et al., 2014; Girshick et al., 2014) . Pretraining offers increased generalization capability especially when the labeled data is scarce and increased inference speed relative to end-to-end training (Devlin et al., 2018). Parallel to our work (Hu et al., 2020; Qiu et al., 2020) proposed pretraining models for GNNs. These contemporary works supervise the models only using attribute or edge generation schemes without accounting for higher order structural similarity or maximizing the mutual information of the embeddings. Recently, (Hu et al., 2019) introduced a framework for pretraining GNNs for graph classification. Different than (Hu et al., 2019) that focuses on graph representations, PanRep aims at node prediction tasks and obtains node representations via capturing properties related to node homophily (Gleich, 2015) or node structural similarity (Rossi & Ahmed, 2014). PanRep is a novel pretrained model for node classification and link prediction that requires significantly less labeled points to reach the performance of its fully supervised counterparts.

## 3 DEFINITIONS AND PROBLEM FORMULATION

A heterogeneous graph with $T$ node types and $R$ relation types is defined as $\mathcal{G} := \{\{\mathcal{V}_t\}_{t=1}^{T}, \{\mathcal{E}_r\}_{r=1}^{R}\}$. The node types represent the different entities and the relation types represent how these entities are semantically associated to each other. For example, in the IMDB network, the node types correspond to actors, directors, movies, etc., whereas the relation types correspond to *directed-by* and *played-in* relations. The number of nodes of type $t$ is denoted by $N_t$ and its associated nodal set by $\mathcal{V}_t := \{n_t\}_{n=1}^{N_t}$. The total number of nodes in $\mathcal{G}$ is $N := \sum_{t=1}^{t} N_t$. The $r$th relation type, $\mathcal{E}_r := \{(n_t, n'_{t'}) \in \mathcal{V}_t \times \mathcal{V}_{t'}\}$, holds all interactions of a certain type among $\mathcal{V}_t$ and $\mathcal{V}_{t'}$ and may represent that a movie is *directed-by* a director. Heterogenous graphs are typically used to represent KGs (Wang et al., 2017). Each node $n_t$ is also associated with an $F \times 1$ feature vector $\mathbf{x}_{n_t}$. This feature may be a natural language embedding of the title of a movie. The nodal features are collected in a $N \times F$ matrix $\mathbf{X}$. Note that certain node types may not have features and for these we use an embedding layer to represent their features.

**Unsupervised learning.** Given $\mathcal{G}$ and $\mathbf{X}$, the goal of representation learning is to estimate a function $g$ such that $\mathbf{H} := g(\mathbf{X}, \mathcal{G})$, where $\mathbf{H} \in \mathbb{R}^{N \times D}$ represents the node embeddings and $D$ is the size of the embedding space.Note that in estimating $g$, no labeled information is available.

**Universal representation learning.** The universal representations $\mathbf{H}$ should perform well on different downstream tasks. Different node classification and link prediction tasks may arise by considering different number of training nodes and links and different label types, e.g., occupation label or education level label. Consider $I$ downstream task, for the universal representations $\mathbf{H}$ it holds that

$$\mathcal{L}^{(i)}(f^{(i)}(\mathbf{H}), \mathcal{T}^{(i)}) \leq \epsilon, \ i = 1, \ldots, I, \tag{1}$$

where $\mathcal{L}^{(i)}$, $f^{(i)}$, and $\mathcal{T}^{(i)}$ represent the loss function, learned classifier, and training set (node labels or links) for task $i$, respectively and $\epsilon$ is the largest error for all tasks. The goal of unsupervised universal representation learning is to learn $\mathbf{H}$ such that $\epsilon$ is small. While learning $\mathbf{H}$, PanRep does not have knowledge of $\{\mathcal{L}^{(i)}, f^{(i)}, \mathcal{T}^{(i)}\}_i$. Nevertheless, by utilizing the novel decoder scheme PanRep achieves superior performance even compared to supervised approaches across tasks.

## 4 PANREP

Our universal representation learning framework aims at embedding nodes in a low-dimensional space such that the representations are discriminative for node classification and link prediction.

Methods for learning over graphs typically rely on modeling homophily of nodes that postulates neighboring vertices to have similar attributes (Smola & Kondor, 2003; Yuan et al., 2014) or structural similarity among nodes (Rossi & Ahmed, 2014), where vertices involved in similar graph structural patterns possess related attributes (Donnat et al., 2018). Motivated by these methods we identify related properties encoded in the graph data. Clustering nodes based on their attributes provides a strong signal for node homophily (Koren et al., 2009). Network motifs reveal the local structure information for nodes in the graph (Ahmed et al., 2017). Metapaths encode the heterogeneous graph neighborhood and indicate the local connectivity (Dong et al., 2017). Finally, maximizing the mutual information among embeddings declusters node representations and provides further discriminative information (Veličković et al., 2018b).

## 4.1 Universal supervision signals

In order to capture the aforementioned properties we develop four novel universal decoders.

**Cluster and recover supervision.** Node attributes may reveal interesting properties of nodes, such as clusters of customers based on their buying power and age. This is important in recommendation systems, where traditional matrix factorization approaches (Koren et al., 2009) rely on revealing clusters of similar buyers. To capitalize such information we propose to supervise the universal embeddings by such cluster representations. Specifically, we cluster the node attributes via $K$-means (Kanungo et al., 2002) and then design a model that decodes $\mathbf{H}$ to recover the original clusters. The CR-decoder is modeled as a two layer MLP and is supervised by

$$\mathcal{L}_{\mathrm{CR}} := -\sum_{n=1}^{N}\sum_{k=1}^{K} C_{nk} \ln \hat{C}_{nk}, \tag{2}$$

where the cluster assignment $C_{nk}$ is 1 if node $n$ belongs in class $k$ and the predicted cluster assignment $\hat{C}_{nk}$ is the output of the CR-decoder. We showcase in the appendix that this signal is superior to the attribute generation scheme used in (Hu et al., 2020). Such a supervision signal will enrich the universal embeddings $\mathbf{H}$ with information based on the clustering of local node features.

**Motif supervision.** Network motifs are sub-graphs where the nodes have specific connectivity patterns. Typical size-3 motifs for example, are the triangle and the star motifs. Each of these sub-graphs is identified by a particular pattern of interactions among nodes, and reveals important properties for the participating nodes. In gene regulatory networks for example, motifs are associated with certain biological properties (Babu et al., 2004). The work in (Ahmed et al., 2017) develops efficient parallel implementations for extracting network motifs. We aspire to capture structural similarity among nodes by predicting their motif information. The motivation is that nodes which might be distant in the graph may have similar structural properties as described by their motifs.

Using the method in (Ahmed et al., 2017) we extract a frequency vector $\boldsymbol{\mu}_n$ per node that shows how many times $n$ participates to graph motifs up to size 4. This information reveals the structural role of nodes such as star-center, star-edge nodes, or bridge nodes. The motif decoder predicts this vector for all nodes using the universal representation $\mathbf{H}$. This allows for information sharing among nodes which are far away in the graph but have similar motif frequency vectors. The novel motif decoder is modeled as a two-layer MLP and is supervised by the following loss function

$$\mathcal{L}_{\mathrm{MOT}} := \sum_{n=1}^{N} \|\boldsymbol{\mu}_n - \hat{\boldsymbol{\mu}}_n\|_2^2 \tag{3}$$

where $\hat{\boldsymbol{\mu}}_n$ is the output of the Mot-decoder for the $n$th node. Using the Mot-decoder PanRep enhances the universal embeddings by structural information encoded in the node motifs.

**Metapath RW supervision.** Metapaths are sequences of edges of possibly different type that connect nodes in a KG. A metapath random walk (MRW) is a specialized RW that follows different edge-types (Dong et al., 2017).

We aspire to capture local connectivity patterns by promoting nodes participating in a MRW to have similar embeddings. Consider all node pairs for nodes $(n_t, n'_{t'})$ participating in a MRW, the following criterion maximizes the similarity among these nodes as follows

$$\mathcal{L}_{\mathrm{MRW}} := \log(1 + \exp(-y \times \mathbf{h}_{n_t}^{\top}\mathrm{diag}(\mathbf{r}_{t,t'})\mathbf{h}_{n'_{t'}})), \tag{4}$$

where $\mathbf{h}_{n_t}$ and $\mathbf{h}_{n'_{t'}}$ are the universal embeddings for nodes $n_t$ and $n'_{t'}$, respectively, $\mathbf{r}_{t,t'}$ is an embedding parametrized on the pair of node-types and $y$ is 1 if $n_t$ and $n'_{t'}$ co-occur in the MRW and

-1 otherwise. Negative examples are generated by randomly selecting tail nodes for a fixed head node with ratio 5 negatives per positive example. Link prediction is indeed a special case of the MRW supervision that considers MRWs of length 1. However, metapaths convey more information than regular links since the former can be defined to promote certain prior knowledge. For example, in predicting the movie genre in IMDB the metapath configured by the edge types (played by, played in) among node types (movie, actor, movie) will potentially connect movies with same genre and hence it is desirable. The embedding per node-type pair $\mathbf{r}_{t,t'}$ allows the MRW-decoder to weight the similarity among node embeddings from different node types accordingly. The length of the MRW controls the radius of the graph neighborhood considered in equation 4 and it can vary from local to intermediate.

**Heterogenous information maximization.** The aforementioned supervision signals capture clustering affinity, structural similarity and local and intermediate neighborhood of the nodes. Nevertheless, further information can be extracted by the representations by maximizing the mutual information among node representations. Such an approach for homogeneous graphs is detailed in (Veličković et al., 2018b), where the mutual information between node representations and the global graph summaries is maximized (Hjelm et al., 2018).

Towards further refining the universal embeddings, we propose a generalization of (Veličković et al., 2018b) for heterogeneous graphs. We consider a global summary vector per $t$ as $\mathbf{s}_t := \sum_{n_t=1}^{N_t} \mathbf{h}_{n_t}$ that captures the average $t$th node representation. We aspire to maximize the mutual information among $\mathbf{s}_t$ and the corresponding nodes in $\mathcal{V}_t$. The proposed HIM decoder is supervised by the following contrastive loss function

$$\mathcal{L}_{\text{HIM}} := \sum_{t=1}^{T} \left( \sum_{n_t=1}^{N_t} \log \left( \sigma(\mathbf{h}_{n_t}^{\top} \mathbf{W} \mathbf{s}_t) \right) + \log \left( 1 - \sigma(\tilde{\mathbf{h}}_{n_t}^{\top} \mathbf{W} \mathbf{s}_t) \right) \right) \tag{5}$$

where the bilinear scoring function $\sigma(\tilde{\mathbf{h}}_{n_t}^{\top} \mathbf{W} \mathbf{s}_t)$ captures how close is $\mathbf{h}_{n_t}$ to the global summary, $\mathbf{W}$ is a learnable matrix and $\tilde{\mathbf{h}}_{n_t}$ represents the negative example used to facilitate training. Designing negative examples is a cornerstone property for training contrastive models (Veličković et al., 2018b). We generate the negative examples in 5 by shuffling node attributes among nodes of the same type. The HIM decoder maximizes the mutual information across nodes and complements the former decoders.

**Putting everything together.** PanRep's final loss function is the linear combination of 2-5 and can be considered in the framework of deep multitask learning (Zhang et al., 2014), since the GNN encoder is shared across the multiple supervision tasks and PanRep makes multiple inferences in one forward pass. Such networks are not only scalable, but the shared features within these networks can induce more robust regularization and possibly boost performance (Chen et al., 2017). Introducing adaptive weights per decoder to control its learning rate is a future direction of PanRep.

### 4.2 PanRep Encoder

Although the PanRep framework can utilize any GNN model as an encoder (Wu et al., 2020), in this paper PanRep uses a relational (R)GCN encoder (Schlichtkrull et al., 2018). RGCNs extend the graph convolution operation (Kipf & Welling, 2017) to heterogenous graphs. The $l$th RGCN layer computes the $n$th node representation $\mathbf{h}_n^{(l+1)}$ as follows

$$\mathbf{h}_n^{(l+1)} := \sigma \left( \sum_{r=1}^{R} \sum_{n' \in \mathcal{N}_n^r} \mathbf{h}_{n'}^{(l)} \mathbf{W}_r^{(l)} \right), \tag{6}$$

where $\mathcal{N}_n^r$ is the neighborhood of node $n$ under relation $r$, $\sigma$ the rectified linear unit non linear function, and $\mathbf{W}_r^{(l)}$ is a learnable matrix associated with the $r$th relation. Essentially, the output of the RGCN layer for node $n$ is a nonlinear combination of the hidden representations of neighboring nodes weighted based on the relation type.

**Alternative encoders.** Several works consider designing possibly more general GNN encoders that utilize attention mechanism (Wang et al., 2019b; Veličković et al., 2018a) or graph isomorphism networks (Xu et al., 2019). This paper proposes novel supervision signals for unsupervised learning that capture general properties of the graph data. Designing a universal encoder based on these contemporary GNN models is an interesting future direction of PanRep.

Table 1: Node classification results.

| | Train % | | Unsupervised | | | | Semi-supervised | | | |
|---|---|---|---|---|---|---|---|---|---|---|
| | | | node2vec | meta2vec | HIM | PanRep | HAN | MAGNN | RGCN | PanRep-FT |
| IMDB | Mac-F1 | 40% | 50.63 | 47.57 | 55.21 | **56.04** | 56.15 | **60.27** | 58.48 | 59.49 |
| | | 60% | 51.65 | 48.17 | 57.66 | **58.51** | 57.29 | **60.66** | 58.42 | 59.86 |
| | | 80% | 51.49 | 49.99 | 57.89 | 60.23 | 58.51 | 61.44 | 58.76 | **61.49** |
| | Mic-F1 | 40% | 51.77 | 48.17 | 55.11 | **55.92** | 57.32 | **60.50** | 58.64 | 59.67 |
| | | 60% | 52.79 | 49.87 | 56.57 | **58.41** | 58.42 | **60.88** | 58.55 | 59.75 |
| | | 80% | 52.72 | 50.50 | 57.79 | 60.14 | 59.24 | 61.53 | 58.89 | **61.59** |
| OAG | Mac-F1 | 40% | 56.37 | **65.75** | 50.54 | 57.76 | 63.99 | 63.31 | 64.68 | **64.72** |
| | | 60% | 57.01 | **66.09** | 51.98 | 59.72 | 64.25 | 63.42 | 65.96 | **66.99** |
| | | 80% | 58.05 | **65.75** | 53.25 | 63.03 | 64.37 | 63.89 | 67.67 | **67.90** |
| | Mic-F1 | 40% | 70.17 | 74.54 | 71.91 | **75.50** | 73.95 | 72.74 | **81.92** | 80.36 |
| | | 60% | 70.95 | 74.96 | 73.89 | **77.39** | 75.32 | 72.75 | 81.39 | **81.78** |
| | | 80% | 72.24 | 74.73 | 75.31 | **79.76** | 75.24 | 73.43 | 82.38 | **83.17** |

## 4.3 PANREP-FT

In certain cases a very small subset of labels may be known a priori for the downstream task. In such cases it is beneficial to fine-tune PanRep's model to obtain refined node representations. In this context, PanRep can be considered as pretrained model and a downstream task specific loss may be applied to supervise PanRep. BERT models in natural language processing have reported state of the art results by considering such a pretrain and fine-tune framework (Devlin et al., 2018). PanRep-FT can be considered a counterpart of BERT for extracting information from heterogenous graph data. PanRep-FT combines the benefit of universal unsupervised learning and task specific information and achieves greater generalization capacity especially when labeled data are scarce (Erhan et al., 2010).

## 5 EXPERIMENTS

The proposed universal representation learning techniques are compared with state-of-the-art methods. For node classification the following contemporary methods are considered RGCN (Schlichtkrull et al., 2018), HAN (Wang et al., 2019b), MAGNN (Fu et al., 2020), node2vec (Grover & Leskovec, 2016), meta2vec (Dong et al., 2017) and an adaptations of the work in (Veličković et al., 2018b) for heterogenous graphs termed HIM. For link prediction the baseline models is RGCN (Schlichtkrull et al., 2018) with DistMult supervision (Yang et al., 2014) that uses the same encoder as PanRep. The Mot-decoder and RC-decoder employ a 2-layer MLP. For the MRW-decoder we use length-2 MRWs. The parameters for all methods considered are optimized using the performance on the validation set.

**Datasets.** We consider a subset of IMDB dataset (IMD, 2020) containing 11,616 nodes with 3 node-types and 17,106 edges from 6 edge-types. Each movie is associated with a label representing its genre and with a feature vector capturing its keywords. We also use a subset of the OAG dataset (OAG, 2020) with 23,696 with 4 node types (authors, affiliations, papers, venues) and 90,183 edges from 14 edge-types. In OAG we did not use MOT supervision since (Ahmed et al., 2017) is not applicable. Each paper is associated with a label denoting the scientific area and with an embedding of the papers' text. Finally, we utilize the DRKG constructed in (Ioannidis et al., 2020). DRKG contains 5,874,261 biological interactions belonging to 107 edge-types among 97,238 biological entities from 13 entity-types. For further details on the datasets and configuration of methods see the appendix.

**Scalability.** PanRep is implemented using Deep Graph Library (DGL)'s mini-batch training, which is scalable w.r.t. time and space. During mini-batch training a subset of seed nodes is sampled along with the $K$-hop neighbors of the seed nodes, where $K$ is the depth of the GNN. Furthermore, the number of edges at each level is bounded by using neighbor sampling. This allows training on graphs with billions of nodes/edges since only a fraction of the nodes/edges will be loaded on the GPU per mini-batch. PanRep's complexity is controlled by the RGCN encoder whose computation complexity of a mini-batch is $\mathcal{O}(BF^K)$ for sparse graphs, where $B$ is the batch size, $K$ is the number of layers and $F$ is the maximum number of neighbors per node. By using neighbor sampling, $F$ is usually less than 10–20 and in our experiments we found that K=2 often works well. The decoders' complexity is significantly lower and do not add but a constant factor to the overal complexity and hence the overall per-epoch runtime is similar to that of RGCN.

## 5.1 NODE CLASSIFICATION

In this experiment we compare supervised and unsupervised methods for classification. First, the labeled nodes are split in 10% training, 5% validation, and 85% testing sets. The supervised methods utilize the training labels whereas the unsupervised methods do not use them in calculating the node embeddings. Then the embeddings corresponding to the 85% testing nodes are further split to training

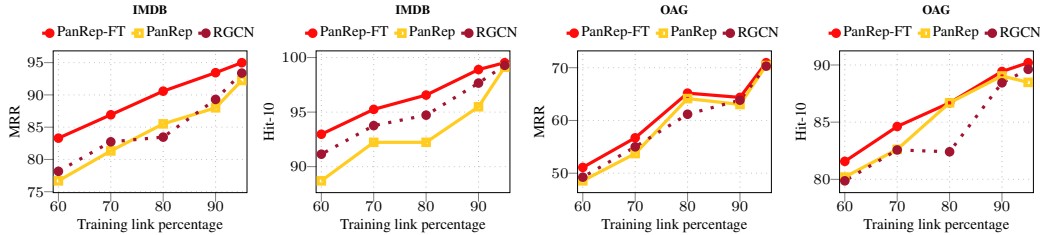

Figure 2: MRR and Hit-10 for link prediction across different percentages of testing links.

Table 2: Node classification results for different labeled supervision splits.

| Datasets | Metrics | Train embeddings % | | 5% | | 10% | | 20% | |
|---|---|---|---|---|---|---|---|---|---|
| | | Train% | PanRep | RGCN | PanRep-FT | RGCN | PanRep-FT | RGCN | PanRep-FT |
| IMDB | Macro-F1 | 40% | 56.04 | 55.12 | **56.85** | 58.48 | **59.49** | 61.30 | **63.14** |
| | | 60% | 58.51 | 55.20 | **59.39** | 58.42 | **59.86** | 60.98 | **62.91** |
| | | 80% | 60.23 | 55.55 | **61.27** | 58.76 | **61.49** | 61.10 | **62.72** |
| | Micro-F1 | 40% | 55.92 | 55.27 | **56.92** | 58.64 | **59.67** | 61.49 | **63.17** |
| | | 60% | 58.41 | 55.39 | **59.45** | 58.55 | **59.75** | 61.17 | **62.89** |
| | | 80% | 60.14 | 55.62 | **61.32** | 58.89 | **61.39** | 61.30 | **62.75** |
| OAG | Macro-F1 | 40% | 57.76 | 55.51 | **64.99** | 64.68 | **64.72** | 67.07 | 65.31 |
| | | 60% | 59.72 | 55.99 | **66.62** | 65.96 | **66.99** | 67.58 | 66.25 |
| | | 80% | 63.03 | 56.36 | **68.94** | 66.10 | **68.60** | 67.67 | **67.90** |
| | Micro-F1 | 40% | 75.50 | 78.00 | **80.19** | 81.92 | 80.36 | 82.57 | 81.17 |
| | | 60% | 77.39 | 78.07 | **81.36** | 81.39 | **81.78** | 81.74 | 81.34 |
| | | 80% | 79.76 | 78.44 | **82.52** | 82.38 | **83.17** | 82.20 | 82.31 |

and testing sets and a linear SVM is trained. This evaluation setting allows us to directly compare the different supervised and unsupervised approaches.

We report the Macro and Micro F1 accuracy for different training percentages of the 85% nodes fed to the SVM classifier in Table 1. It is observed that PanRep outperforms significantly other unsupervised approaches as well as some supervised approaches. In certain splits, PanRep outperforms its supervised counterpart RGCN that uses node labels for supervision. Metapath2vec (Dong et al., 2017) reports competitive performance for OAG in Macro-F1 score but underperformed in Micro-F1. Nevertheless, in this experiment Metapath2vec only uses the best performing metapath that is *paper-venue-paper*, which is considerably better than that of most other metapaths. This way of selecting the metapath, gives Meta2vec an (unfair) advantage over PanRep, which uses all metapaths of length 2 as it computes a universal representation (along with the other supervision signals). PanRep-FT outperforms significantly RGCN that uses the same encoder, which is a testament to the power of pretraining models. Finally, PanRep-FT matches and outperforms in certain splits the state-of-the-art MAGNN that uses a more expressive encoder. PanRep's universal decoders enhance the embeddings with additional discriminative power that results to improved performance in the downstream tasks.

Table 2 reports the accuracy of the PanRep-FT and the encoder RGCN, which is trained directly for the semi-supervised learning task to obtain the embeddings. PanRep-FT consistently outperforms RGCN across most SVM splits, whereas PanRep-FT's advantage over RGCN decreases as more training data are available, which is justifiable since more labels diminish the advantage of pretraining. Moreover, PanRep-FT performs only 100 finetuning epochs with labeled supervision, which is significantly less to the 800 epochs of labeled supervision by RGCN. Nevertheless, PanRep outperforms the supervised RGCN embeddings for 5% training labels. This demonstrates the importance of using PanRep as a pretraining method. RGCN reports similar performance across SVM splits, whereas PanRep-FT increases with more supervision. These results suggest that PanRep-FT's embeddings have higher generalization capacity.

Table 3 reports an ablation study by using different decoder subsets. PanRep that uses all signals obtains the best performance. The decoders HIM and MRW and the their combination exhibit the second best performance. Nevertheless, the full supervision in PanRep leads to the superior performance. Section B in the appendix includes additional experiments.

## 5.2 LINK PREDICTION

Our universal embedding framework is further evaluated for link prediction using the IMDB and OAG datasets. The MRW decoder is used to evaluate the performance of PanRep in link prediction. Figure 2 reports the MRR, and Hit-10 scores of the baseline methods along with the PanRep and PanRep-FT methods. We report the performance of the methods for different percentages of links used for training. Observe that PanRep-FT consistently outperforms the competing methods and

Table 3: Ablation study for different supervision signals.

| Datasets | Metrics | Train % | HIM | MRW | HIM+MRW | MOT | CR | PanRep |
|---|---|---|---|---|---|---|---|---|
| IMDB | Macro-F1 | 40% | 55.21 | 54.54 | 55.32 | 42.28 | 32.21 | **56.04** |
| | | 60% | 57.66 | 56.12 | 57.24 | 43.41 | 35.16 | **58.51** |
| | | 80% | 57.89 | 56.64 | 57.74 | 44.31 | 35.66 | **60.23** |
| | Micro-F1 | 40% | 55.11 | 54.36 | 55.53 | 43.66 | 39.13 | **55.92** |
| | | 60% | 56.57 | 55.91 | 56.25 | 44.89 | 40.03 | **58.41** |
| | | 80% | 57.79 | 56.49 | 58.42 | 45.65 | 40.66 | **60.14** |
| OAG | Macro-F1 | 40% | 50.54 | 55.92 | 56.11 | - | 15.46 | **57.76** |
| | | 60% | 51.98 | 58.40 | 58.91 | - | 15.48 | **59.72** |
| | | 80% | 53.25 | 60.61 | 61.74 | - | 15.54 | **63.03** |
| | Micro-F1 | 40% | 71.91 | 74.39 | 74.65 | - | 63.06 | **75.50** |
| | | 60% | 73.89 | 76.76 | 76.33 | - | 63.14 | **77.39** |
| | | 80% | 75.31 | 78.90 | 78.71 | - | 63.59 | **79.76** |

Table 4: Drug inhibits gene scores for Covid-19.

| PanRep-FT | | | | | RGCN | | | |
|---|---|---|---|---|---|---|---|---|
| Drug name | # hits | Drug name | # hits | | Drug name | # hits | Drug name | # hits |
| Losartan | 232 | Thalidomide | 41 | | Chloroquine | 69 | Tofacitinib | 33 |
| Chloroquine | 198 | Hydroxychloroquine | 19 | | Colchicine | 41 | Ribavirin | 32 |
| Deferoxamine | 104 | Tetrandrine | 13 | | Tetrandrine | 40 | Methylprednisolone | 30 |
| Ribavirin | 101 | Eculizumab | 10 | | Oseltamivir | 37 | Deferoxamine | 30 |
| Methylprednisolone | 44 | Tocilizumab | 9 | | Azithromycin | 36 | Thalidomide | 25 |

We retain the top-10 drugs based on their number of hits for each method. Note that a random classifier will result to approximately 5.3 per drug. This suggests that the reported predictions are significantly better than random.

the performance gain increases as the percentage of training links decreases. This corroborates the advantage of pretraining GNNs for link prediction. Note that PanRep reports similar performance with RGCN that is trained solely in link prediction. This result confirms the success of the universal embeddings in link prediction.

## 5.3 DRUG REPURPOSING

Drug-repurposing aims at discovering the most effective existing drugs to treat a certain disease. Using the Drug Repurposing Knowledge Graph (DRKG) (Ioannidis et al., 2020), we compare the drug repurposing results in Covid-19 among PanRep-FT that is finetuned in link prediction and the baseline RGCN. We employ $L = 1$ hidden layer with $D = 600$ and train for 800 epochs both networks. Drug-repurposing can be formulated as predicting direct links in the DRKG. Here, we attempt to predict whether a drug inhibits a certain gene, which is related to the target disease. We identify 442 genes that are related with the Covid-19 disease. We select 8,104 FDA-approved drugs in the DRKG as candidates; see also the appendix for further information in this experiment. To validate our predictions we use 32 Covid-19 clinical trial drugs. For each gene node we calculate with RGCN and PanRep-FT an inhibit link score associated with every drug. Next, we score all 'drug-inhibits-gene' triples and rank them per target gene. We obtain in this way 442 ranked lists of drugs, one per gene node. Finally, to assess whether our prediction is in par with the drugs used for treatment, we check the overlap among the top 100 predicted drugs and the drugs used in clinical trials per gene. Table 4 lists the clinical drugs included in the top-100 predicted drugs across all the genes with their corresponding number of hits for the RGCN and PanRep-FT. It can be observed, that several of the widely used drugs in clinical trials appear high on the predicted list in both prediction. Furthermore, PanRep-FT reports a higher hit rate than RGCN, which corroborates the benefit of using the universal pretraining decoders. The universal representation endows PanRep with increased generalization power that allows for accurate link prediction performance when training data are extremely scarce as is the case of Covid-19. While this study, does not recommend specific drugs, it demonstrates a powerful deep learning methodology to prioritize existing drugs for further investigation, which holds the potential of accelerating therapeutic development for Covid-19.

## 6 CONCLUSION

This paper develops a novel framework for unsupervised learning of universal node representations on heterogenous graphs termed. PanRep supervises the GNN encoder by decoders attuned to model the clustering of local node features, structural similarity among nodes, the local and intermediate neighborhood structure, and the mutual information among same-type nodes. To further facilitate cases where limited labels are available we implement PanRep-FT. Experiments in node classification and link prediction corroborate the competitive performance of the learned universal node representations compared to unsupervised and supervised methods. Experiments on the DRKG showcase the advantage of the universal embeddings in drug repurposing.

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

Table 5: Attribute generation supervision against clustering and recover supervision for different number of clusters $K$ in the node classification task.

| Datasets | Metrics | Train % | ATTRIBUTE GENERATION | CR $K = 2$ | CR $K = 4$ | CR $K = 5$ | CR $K = 10$ |
|---|---|---|---|---|---|---|---|
| IMDB | Macro-F1 | 40% | 17.52 | 29.52 | 31.94 | **32.21** | 31.22 |
| | | 60% | 17.53 | 30.75 | 34.50 | **35.16** | 34.19 |
| | | 80% | 17.30 | 3145 | 35.02 | **35.66** | 34.82 |
| | Micro-F1 | 40% | 35.66 | 38.51 | 39.40 | **39.13** | 38.38 |
| | | 60% | 35.70 | 38.83 | 41.02 | **41.13** | 40.03 |
| | | 80% | 35.07 | 39.54 | 41.16 | **41.39** | 40.66 |

## A  IMPLEMENTATION FRAMEWORK

The methods presented in this paper are implemented in the efficient deep graph learning (DGL)[2] library Wang et al. (2019a). PanRep is implemented using the mini-batch training framework that facilitates training for very large graphs even with limited computational resources [3]. The competing methods RGCN, MAGNN and HAN are also implemented using the DGL. PanRep experiments are executed on an AWS P3.8xlarge[4] instances with 8 GPUs each having 16GB of memory.

## B  CLUSTER AND RECOVER COMPARED TO NODE ATTRIBUTE GENERATION

In this section we compare the cluster and recover supervsion signal to the attribute generation supervision that is employed by several GNN pretraining methods. In this experiment we employ the setting of Section 5.1 for node classification. The attribute generation supervision employs a 2-layer MLP network that attempts to reconstruct the original nodal attributes given the node embeddings. In Table 5 we report the node classification accuracy for different number of clusters $K$ considered in the CR supervision. It can be observed that the CR methods significantly outperform the attribute generation method. One reason for this is that the attributes are already considered by the GNN encoder in calculating the embeddings, and this supervision does not increase the predictive capacity of the model. On the other hand the clustering supervision allows the GNN to discover clusters of nodes with similar attributes and hence refines the learned embeddings.

## C  METHODS

Different competing methods include RGCN Schlichtkrull et al. (2018), HAN Wang et al. (2019b), MAGNN Fu et al. (2020), node2vec Grover & Leskovec (2016), meta2vec Dong et al. (2017) and an adaptation of the work in Veličković et al. (2018b) for heterogenous graphs termed HIM. For link prediction the baseline model is RGCN Schlichtkrull et al. (2018) with DistMult supervision Yang et al. (2014) that uses the same encoder as PanRep. The Mot-decoder and RC-decoder employ a 2-layer MLP. For the MRW-decoder we use length-2 MRWs. The parameters for all methods considered are optimized using the performance on the validation set. For the majority of the experiments PanRep uses a hidden dimension of 300, 1 hidden layer, 800 epochs of model training, 100 epochs for finetuning, and an ADAM optimzer Kingma & Ba (2015) with a learning rate of 0.001. For link prediction finetuning PanRep uses a DistMult model Yang et al. (2014) whereas for node classification it uses a logistic loss.

## D  DRUG REPURPOSING EVALUATION

Our work focused entirely on *evaluating* the ranked list of candidate drugs against a set of 32 drugs that, at the time of submission, were in Covid-19 clinical trials (Gordon et al., 2020; Zhou et al., 2020). Specifically, given a gene whose associated protein is a potential drug target, we used the approach described in section 5.3 to rank the 8104 FDA approved drugs and select the 100 highest ranked drugs. We evaluated the performance of a method by looking at the intersection between these 100 highest ranked drugs and the 32 drugs in Covid-19 trials. In our experiments, we used 442 genes that researchers have identified as potential targets for Covid-19 and computed 442 ranked lists and associated intersections. We aggregated the results by counting the number of times each one of the 32 drugs were in these intersections, and the drugs with the highest counts are shown in Table 4.

---

[2]https://www.dgl.ai/
[3]https://github.com/dmlc/dgl/blob/master/examples/pytorch/rgcn-hetero
[4]https://aws.amazon.com/ec2/instance-types/p3/

Table 6: Statistics of datasets.

| Dataset | Node | Edge |
|---|---|---|
| IMDb | # movie (M): 4,278
# director (D): 2,081
# actor (A): 5,257 | # M-directed by-D: 4,278 , D-directed-M: 4,278
# M-played by-A: 12,828, A-played-M: 12,828 |
| OAG | # author (A): 13,720
# paper (P): 7,326
# affiliation (Af): 2,290
# venue (V): 782 | # P-in journal-V: 3941, V-journal has-P: 3941
# P-conference-V: 3368, V-conference has-P: 3368
# P-cites-P: 3327, P-cited by-P: 3327
# A-writes as last-P: 4522, P-written by last-A: 4522
# A-writes as other-P: 7769, P-written by other-A: 7769
# A-writes as first-P: 4795, P-written by first-A: 4795
# A-affiliated with-Af: 17035, Af-affiliated with-A: 17035 |

Table 7: Number of nodes per node type in the DRKG and the data sources.

| Entity type | Drugbank | GNBR | Hetionet | STRING | IntAct | DGIdb | Bibliography | Total Entities |
|---|---|---|---|---|---|---|---|---|
| Anatomy | - | - | 400 | - | - | - | - | 400 |
| Atc | 4,048 | - | - | - | - | - | - | 4,048 |
| Biological Process | - | - | 11,381 | - | - | - | - | 11,381 |
| Cellular Component | - | - | 1,391 | - | - | - | - | 1,391 |
| Compound | 9,708 | 11,961 | 1,538 | - | 153 | 6,348 | 6,250 | 24,313 |
| Disease | 1,182 | 4,746 | 257 | - | - | - | 33 | 5,103 |
| Gene | 4,973 | 27,111 | 19,145 | 18,316 | 16,321 | 2,551 | 3,181 | 39,220 |
| Molecular Function | - | - | 2,884 | - | - | - | - | 2,884 |
| Pathway | - | - | 1,822 | - | - | - | - | 1,822 |
| Pharmacologic Class | - | - | 345 | - | - | - | - | 345 |
| Side Effect | - | - | 5,701 | - | - | - | - | 5,701 |
| Symptom | - | - | 415 | - | - | - | - | 415 |
| Tax | - | 215 | - | - | - | - | - | 215 |
| Total | 19,911 | 44,033 | 45,279 | 18,316 | 16,474 | 8,899 | 9,464 | 97,238 |

# E    DATASETS

## E.1    DRKG

The *Drug Repurposing Knowledge Graph* (DRKG) contains 97055 entities belonging to 13 entity-types Ioannidis et al. (2020). The type-wise distribution of the entities is shown in Table 7. DRKG contains a total of 5869294 triplets belonging to 107 edge-types. Table 8 shows the number of triplets between different entity-type pairs for DRKG and various data sources. The DRKG is publicly available.[5]

## E.2    IMDB AND OAG

IDMB IMD (2020) is a movie database including information about the cast, production crew, and plot summaries. A subset of IMDb is used after data preprocessing in Table 6. Movies are labeled as one of three classes (Action, Comedy, and Drama) based on their genre information. Each movie is also described by a bag-of-words representation of its plot keywords.

OAG OAG (2020) is bibliography website. We preprocess the data and retain the subgraph in Table 6. The papers are divided into 6 research areas. Each paper is described by a BERT embedding of the paper's title.

---

[5]https://github.com/gnn4dr/DRKG/

Table 8: Number of interactions in the DRKG and the data sources.

| Entity-type pair | Drugbank | GNBR | Hetionet | STRING | IntAct | DGIdb | Bibliography | Total interactions |
|---|---|---|---|---|---|---|---|---|
| ('Gene', 'Gene') | - | 667, 22 | 474, 526 | 1, 496, 708 | 254, 346 | - | 58, 629 | 2, 350, 931 |
| ('Compound', 'Gene') | 24, 801 | 80, 803 | 51, 429 | - | 1, 805 | 26, 290 | 25, 666 | 210, 794 |
| ('Disease', 'Gene') | - | 95, 399 | 27, 977 | - | - | - | 461 | 123, 837 |
| ('Atc', 'Compound') | 15, 750 | - | - | - | - | - | - | 15, 750 |
| ('Compound', 'Compound') | 1, 379, 271 | - | 6, 486 | - | - | - | - | 1, 385, 757 |
| ('Compound', 'Disease') | 4, 968 | 77, 782 | 1, 145 | - | - | - | - | 83, 895 |
| ('Gene', 'Tax') | - | 14, 663 | - | - | - | - | - | 14, 663 |
| ('Biological Process', 'Gene') | - | - | 559, 504 | - | - | - | - | 559, 504 |
| ('Disease', 'Symptom') | - | - | 3, 357 | - | - | - | - | 3, 357 |
| ('Anatomy', 'Disease') | - | - | 3, 602 | - | - | - | - | 3, 602 |
| ('Disease', 'Disease') | - | - | 543 | - | - | - | - | 543 |
| ('Anatomy', 'Gene') | - | - | 726, 495 | - | - | - | - | 726, 495 |
| ('Gene', 'Molecular Function') | - | - | 97, 222 | - | - | - | - | 97, 222 |
| ('Compound', 'Pharmacologic Class') | - | - | 1, 029 | - | - | - | - | 1, 029 |
| ('Cellular Component', 'Gene') | - | - | 73, 566 | - | - | - | - | 73, 566 |
| ('Gene', 'Pathway') | - | - | 84, 372 | - | - | - | - | 84, 372 |
| ('Compound', 'Side Effect') | - | - | 138, 944 | - | - | - | - | 138, 944 |
| Total | 1, 424, 790 | 335, 369 | 2, 250, 197 | 1, 496, 708 | 256, 151 | 26, 290 | 84, 756 | 5, 874, 261 |

