# OpenReview forum: "PanRep: Universal node embeddings for heterogeneous graphs"
_ICLR.cc/2021/Conference — Reject_

### Official Review · AnonReviewer3 · 2020-10-27
**Official Blind Review #3**

**Rating:** 5
**Confidence:** 4

**Review:**

This paper proposed introduces a problem formulation of universal unsupervised learning. They develop an unsupervised node representation learning method by combining four signals: (1) cluster and recover supervision, (2) motif supervision, (3) metapath random walk supervision, and (4) heterogeneous information maximization. Aside from conducting experiments on node classification and link prediction, they apply their method on drop-repurposing knowledge graph to discover drugs for Covid-19.

I agree that there is contribution to putting these supervision signals together and showing results on drug-repurposing for COVID-19. However, the proposed approach is rather a mix of previous works and hence not novel.
- All four supervision signals are based on existing works. What's the novelty in all four signals? Recovering cluster assignments, calculating motif frequency vectors, and proximity-based embedding (i.e. RW) are all well-known methods in the graph domain and the objective used in "Heterogenous information maximization" to me seems nearly identical to the one proposed in Deep Graph Infomax [1]
- The performance improvement can simply a result of loss ensembles. How does each individual supervision signal contribute to the performance? An ablation study discussing that should be conducted.

For the cluster and recover supervision signal, Panrep attempts to recover results obtained from K-Means. However,
- K-Means perform poorly on many datasets (e.g. graphs where true communities are highly overlapping). I doubt that learning to recover such signals would be beneficial.
- how do you decide on the number of communities $K$ and does it affect performance a lot?
- Equation (2)  should be permutation invariant to communities (invariant to the ordering of communities)

[1] Petar Velickovi ˇ c, William Fedus, William L Hamilton, Pietro Liò, Yoshua Bengio, and R Devon ´
Hjelm. Deep graph infomax. arXiv preprint arXiv:1809.10341, 2018b.

---

### Official Review · AnonReviewer2 · 2020-10-28

**Rating:** 5
**Confidence:** 4

**Review:**

Pros:

(1) This paper is well-organized and easy to follow, clearly presenting the composition of the proposed model.

(2) Motivation of this paper is strong, and readers may foresee rich applications.

(3) Each model component is with a clear goal, facilitating the interpretation of the proposed approach.

Cons:

(1) This paper is not technical. All techniques are not originally proposed, and the whole model is simply ‘putting everything together’.

(2) For all ablation studies, no evidence is shown which component contributes to the performance improvement compared with the best single-component performance.

(3) No performance variance is provided to make the improvement more convincing considering the close performances of some cases.

- - - -

Detailed Comments:

This paper proposes an approach for unsupervised learning of universal node representations on heterogenous graphs. Specifically, four components are presented with individual loss functions to ensure the representations encoding a certain category of information, and the final model is attained by ‘putting everything together’. Overall, this paper is well-organized and easy to follow, clearly presenting a well-motivated model and how each component handles the information extraction. We can also foresee great application value with the improved fitting ability on universal node representations. However, all techniques are not originally proposed, making this paper less technical. Besides, no evidence is shown which component contributes to the performance improvement compared with the best single-component performance in the ablation study. Also, no performance variance is provided to make the improvement more convincing considering the close performances of some cases in the experiments. To make the paper more convincing, I would have the following suggestions:

(1) Ablation study should be enriched. For the current ablation study, only one component contributes to the main performance. Both the specification of where the performance improvement comes from and how components cooperation facilitates the performance are of little evidence.

(2) More details should be provided on why larger training datasets (e.g., from 40% to 80%) can degrade the performance of some baselines.

(3) For experiments, variance or standard deviation should be provided to make the minor improvements in some cases convincing. Also, performance on more diverse datasets should also be presented for generalization purposes.

(4) In drug-repurposing experiments, the conclusion can also be obtained from Table 4 that PanRep-FT tends to have long-tail effect; however RGCN shows more stable performance. To better evaluate the performance, it is better to present the distribution among all drugs and evaluate the performance of the two models from different perspectives including application consideration.

(5) There is no information on how to balance different loss terms. For example, at least five loss terms should be delicately tuned for a balance in PanRep-FT. However, the author does not present how to balance the weights among loss terms. Besides, I would argue the experimental weights among loss terms should be presented in detail to determine how each component contributes to the final performance.

---

### Official Review · AnonReviewer4 · 2020-10-29
**Interesting paper on universal graph pretraining on heterogeneous graphs**

**Rating:** 6
**Confidence:** 3

**Review:**

Summary: This paper proposes a universal and unsupervised GNN-based representation learning (node embedding pretraining) model named PanRep for heterogeneous graphs, which benefits a variety of downstream tasks such as node classification and link prediction. More specifically, employing an encoder similar to R-GCN, PanRep utilizes four different types of universal supervision signals for heterogeneous graphs, i.e. cluster and recover, motif, metapath random walk and heterogeneous info maximization, to better characterize the graph structures. PanRep can be further fine-tuned in a semi-supervised manner with limited labeled data, known as PanRep-FT, for specific applications. Experiments on benchmark datasets have shown the effectiveness and performance gain over other unsupervised and some supervised baseline approaches. One example use case is applied to COVID-19 drug repurposing to identify potential treatment candidates.

Strong aspects:
(1) The motivation, problem formulation, and model illustration are well explained.
(2) Proposed four types of universal supervision signals are interesting and generalizable for all (heterogeneous) graphs.
(3) Extensive experiments, ablation study, and case study are supportive.

Weak aspects:
(1) Some comparable SOTA baselines may not be considered.
(2) The heterogeneous information maximization signal seems not clearly explained. (See detailed comments below)

Detailed comments:
(1) It is strongly recommended in related work to explicitly explain the difference from other graph pretraining models such as [1,2] and the reason why they are excluded from baselines. Though some work (partially) focus on graph-level application, some node-level pretraining methods are generally comparable and applicable for heterogeneous graph representation learning.

(2) Regarding the drug repurposing task, some baseline approaches (TransE, etc) implemented by DLG-KE on DKRG dataset are not reported [4]. It would be more beneficial and convincing to include the results as well by comparing the #hit other than R-GCN.
(3) HIM is an important component from the observations in the ablation study, it requires a clearer explanation for better
understanding. Some questions are: For the bilinear scoring function, why one node close to its global summary is the ultimate goal for HIM while other signals are used to learn its structures and the nodes are different from each other even though they are of the same type? Why is it reasonable to design contrastive negative samples as introduced in PanRep? What is the type of information of the nodes are fully not available (for example, the node type is not defined in the ontology)?

References:
[1] Hu, Weihua, et al. "Strategies for Pre-training Graph Neural Networks." ICLR (2020).

[2] Hu, Ziniu, et al. "Heterogeneous graph transformer." Proceedings of The Web Conference 2020. 2020.

[3] Ioannidis, Vassilis N., et al. "Drkg-drug repurposing knowledge graph for covid-19." (2020). Github: https://github.com/gnn4dr/DRKG/tree/master/drug_repurpose

---

### Official Review · AnonReviewer1 · 2020-10-30
**incremental novelty and weak baselines**

**Rating:** 4
**Confidence:** 5

**Review:**

PanRep: Universal node embeddings for heterogeneous graphs

The manuscript proposes PanRep, an unsupervised node embedding learning model for heterogeneous graphs. PanRep considers four unsupervised tasks of cluster, motif, heterogenous structure, and mutual information (contrastive learning). Experiments on two different tasks (i.e., node classification and link prediction) demonstrate that PanRep outperforms some baseline methods.

Pros
1. The problem is interesting and important. It is meaningful to study heterogeneous graph pre-training.
2. The presentation is overall good. The content is clear.
3. Four interesting unsupervised tasks are proposed.

Cons
1. The novelty of PanRep is incremental. There are a number of studies [1,2,3,4,5] for graph pre-training. Despite the difference in unsupervised task design, I did not find exciting thing from this work.

2. Experiments should be improved. Baseline methods are relatively weak, it is better to compare and discuss more recent work for graph pre-training [1,2,3,4].

[1] Strategies for Pre-training Graph Neural Networks, ICLR 2020

[2] GCC: Graph Contrastive Coding for Graph Neural Network Pre-Training, KDD 2020

[3] Gpt-gnn: Generative pre-training of graph neural networks, KDD 2020

[4] Self-supervised Auxiliary Learning with Meta-paths for Heterogeneous Graphs, NeurIPS 2020

[5] Self-supervised Learning on Graphs: Deep Insights and New Directions, ArXiv

To summarize, the novelty of this work is incremental and the experiments could be improved.

---

### Decision · Program_Chairs · 2021-01-07
**Final Decision**

**Decision:**

Reject

**Comment:**

Although the paper is clearly written overall and well motivated, reviewers raised several crucial concerns and, unfortunately, the authors did not respond to reviews.

During the discussion, reviewers agree with that this submission is not ready for publication. In particular, empirical evaluation is not thorough as important baselines are not included and discussion is not convincing.

I will therefore reject the paper.
For future submission, I strongly recommend the authors to do author response. There are many cases where the reviewers change their scores based on the interaction between the authors and the reviewers, which is healthy for the review process.